# Comparison of Random Forest and XGBoost Classifiers Using Integrated Optical and SAR Features for Mapping Urban Impervious Surface

**Zhenfeng Shao, Muhammad Nasar Ahmad \* and Akib Javed**

State Key Laboratory of Information Engineering in Surveying, Mapping and Remote Sensing, Wuhan University, Wuhan 430079, China; shaozhenfeng@whu.edu.cn (Z.S.); akibjaved@whu.edu.cn (A.J.)
\* Correspondence: mnasarahmad@whu.edu.cn

**Abstract:** The integration of optical and SAR datasets through ensemble machine learning models shows promising results in urban remote sensing applications. The integration of multi-sensor datasets enhances the accuracy of information extraction. This research presents a comparison of two ensemble machine learning classifiers (random forest and extreme gradient boost (XGBoost)) classifiers using an integration of optical and SAR features and simple layer stacking (SLS) techniques. Therefore, Sentinel-1 (SAR) and Landsat 8 (optical) datasets were used with SAR textures and enhanced modified indices to extract features for the year 2023. The classification process utilized two machine learning algorithms, random forest and XGBoost, for urban impervious surface extraction. The study focused on three significant East Asian cities with diverse urban dynamics: Jakarta, Manila, and Seoul. This research proposed a novel index called the Normalized Blue Water Index (NBWI), which distinguishes water from other features and was utilized as an optical feature. Results showed an overall accuracy of 81% for UIS classification using XGBoost and 77% with RF while classifying land use land cover into four major classes (water, vegetation, bare soil, and urban impervious). However, the proposed framework with the XGBoost classifier outperformed the RF algorithm and Dynamic World (DW) data product and comparatively showed higher classification accuracy. Still, all three results show poor separability with bare soil class compared to ground truth data. XGBoost outperformed random forest and Dynamic World in classification accuracy, highlighting its potential use in urban remote sensing applications.

**Keywords:** data fusion; impervious surface; Landsat 8; random forest; XGBoost

## 1. Introduction

Accurate and timely mapping of urban land use and land cover is crucial using a single satellite dataset [1]. Therefore, integrating multiple data sources to extract more valuable information than individual sources in remote sensing can provide comparatively higher accuracy [2,3]. The importance of data fusion techniques has grown significantly, playing a crucial role in improving the reliability and interpretability of remotely sensed data. It combines data of different types, formats, spatial and temporal scales, and other characteristics to create a unified view [4]. Data fusion techniques are increasingly used in various applications, including remote sensing [5], robotics, surveillance, and medical science. A key benefit of remote sensing data fusion is that it allows the complementary strengths of different data sources and modalities. The aggregated view from fusion techniques can reduce noise [6], fill gaps, resolve uncertainties [7], and improve overall data quality [8]. Fusing optical and radar satellite data mitigates issues like cloud cover in optical data [9].

In remote sensing, data fusion methods are broadly categorized into image-, feature-, and decision-level fusion [10]. Image-level fusion combines pixel values from different images to create composite images [11]. Feature-level fusion extracts features from

each data source and merges them into a unified representation before classification [12]. Decision-level fusion maintains distinct features, performs separate categorization, and then integrates the decisions [13].

There are several sophisticated optical–SAR (OS) fusion-based methods [14,15] that are available, such as intensity–hue–saturation (IHS) transformation, principal component analysis (PCA), wavelet transform, Gram–Schmidt spectral sharpening, and Brovey transform [16]. In contrast with simple layer stacking (SLS), these methods involve various transformations such as intensity, uncorrelated components, frequency bands, use of panchromatic data, and ratio-based processing [17], respectively. These methods provide benefits like spatial sharpening, multi-resolution analysis, and dimensionality reduction for some particular remote sensing applications. Also, applying sophisticated fusion techniques for urban applications sometimes poses challenges, including computational demands and variations in data consistency [18].

SLS is a simple technique that does not require any data transformation or mathematical manipulation. It involves stacking layers from different sensors, such as SAR and optical, to generate a composite image [19]. Specific research studies have been conducted to explore the effectiveness of SLS with data and methods in improving classification results [20].

Research conducted by [21] applied the simple method of layer stacking for data fusion and found that integrating all spectral and backscattering bands yielded the highest mapping result. An overall classification accuracy of 91.07% was achieved with S1-S2, compared to S-2 only (89.53%) and S-2 with radiometric indexes (89.45%). Ref. [22] mentioned that the integrated layer stack composed of Sentinel radar and VNIR data outperformed (OA 88.75%) when applied through the SVM classification. In comparison, the RF (random forest) algorithm achieved an OA of 55.50%, while the K-nearest neighbor algorithm achieved an OA of 39.37%. The authors in [23] used SAR and simulated Sentinel-2 images and after spatial resampling, the fused image stack, which was obtained through layer stacking, was used by SVM for classification. However, they did not make any comparison with existing products. In another study, Ref. [24] implemented the simple layer stacking to fuse Sentinel-1 and Sentinel-2 data at the pixel level.

The aforementioned studies highlighted the potential of data fusion, fusion types, fusion methods, and their advantages for enhancing classification results and accuracy. Therefore, new approaches are still required to be developed for fusing multiple-sensor datasets at city, regional, and global scales. Therefore, the current research focuses on applying the SLS method using extreme gradient boosting (XGBoost). This study presents a framework based on optical and SAR features-based data fusion using SLS and XGBoost algorithms. The novelty of this research is integrating features extracted from Landsat 8 using modified indices and textures from Sentinel-1 SAR datasets to improve classification accuracy.

In this study, we implemented an innovative framework based on an open-source, available multi-sensor datasets to extract impervious surface information in densely populated urban areas. An additional innovative component in our approach involves integrating SLS with the XGBoost machine learning algorithm, which is aimed at significantly improving classification accuracy, particularly in urban land cover scenarios.

Furthermore, previous studies have explored the applications of data fusion in urban mapping, and there is a notable gap regarding the comparison with existing land cover products. This study additionally aims to validate the obtained results against the established global product DW [25]. It assesses the accuracy of the current approach in selected cities, providing valuable insights for the future application of the research findings.

## 2. Materials and Methods

### 2.1. Proposed Framework

The proposed research framework, presented in Figure 1, integrates optical–SAR data using the SLS technique. The primary objective is to enhance the accuracy of urban

impervious surface (UIS) extraction by combining multi-sensor remote sensing datasets from Sentinel-1 (SAR) and Landsat 8 (optical) sensors.

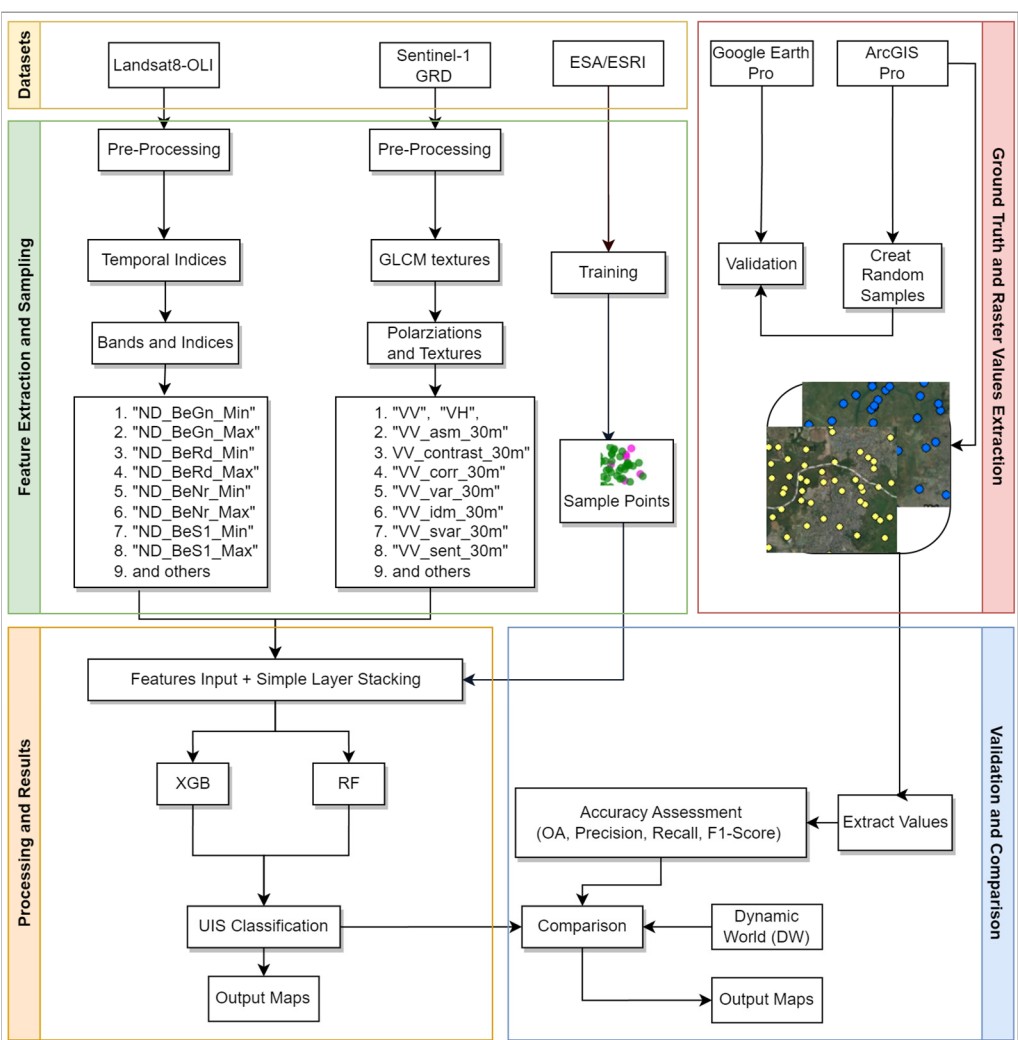

**Figure 1.** Proposed framework.

For Landsat 8 data, a novel index, the Normalized Blue Water Index (NBWI), and modified indices such as the Visible Atmospherically Resistant Index (VARI) and the Normalized Difference Built-up Index (NDBI) were implemented, utilizing all available images for the selected year (2023). This allows for the capture of surface characteristic variations. For Sentinel-1 data, textural features such as local variance, dissimilarity, and entropy were derived from the SAR data to capture spatial patterns and heterogeneity in urban surfaces.

The SLS approach involves stacking multiple layers to capture the spectral characteristics associated with impervious surfaces, including attributes such as color, texture, and reflectance. This technique provides a comprehensive view of impervious surfaces at the city scale. However, implementing advanced data fusion techniques for large-scale applications poses challenges, such as computational requirements and integrating diverse decision-making sources. The study integrates optical–SAR datasets using SLS, stacking the features from both sensors into a composite image. This creates a comprehensive input dataset for the subsequent classification process, which uses RF and XGBoost machine learning classifiers.

The accuracy of UIS extraction is evaluated by comparing the results with existing global data products, specifically the Dynamic World (DW) global data product provided

by Google for 2023. The methodology and materials used in the study are described in detail in the subsequent sub-sections.

### 2.2. Study Area

This research focuses on three cities: Jakarta, Manila, and Seoul. These cities exhibit distinct characteristics regarding climate, population densities, and urbanization rates. Table 1 provides the details of all cities with their climatic zones based on Köppen–Geiger climate classification [26] and population in 2023, according to the World Bank.

**Table 1.** Population and climatic zones of the Selected Cities.

| No. | City Name | Climatic Zone | Population 2023 |
| --- | --- | --- | --- |
| 1 | Jakarta | Tropical | 11,248,839 |
| 2 | Manila | Humid subtropical | 14,667,089 |
| 3 | Seoul | Humid continental with dry winters | 9,988,049 |

Jakarta, Manila, and Seoul are densely populated capital cities in Indonesia, the Philippines, and South Korea. Each city, with its unique geographical features and urban landscapes, presents distinct challenges and opportunities. Jakarta is experiencing rapid urban expansion, Manila is a vibrant and dynamic cityscape serving as the country center region, and Seoul is known for its modern and cosmopolitan environment. Studying impervious surfaces in these cities can provide valuable insights for urban planning, environmental management, and resilience, including land use management, green infrastructure design, mitigation of urban flooding [27], and water quality impacts [28,29]. Their geographical locations are shown in Figure 2.

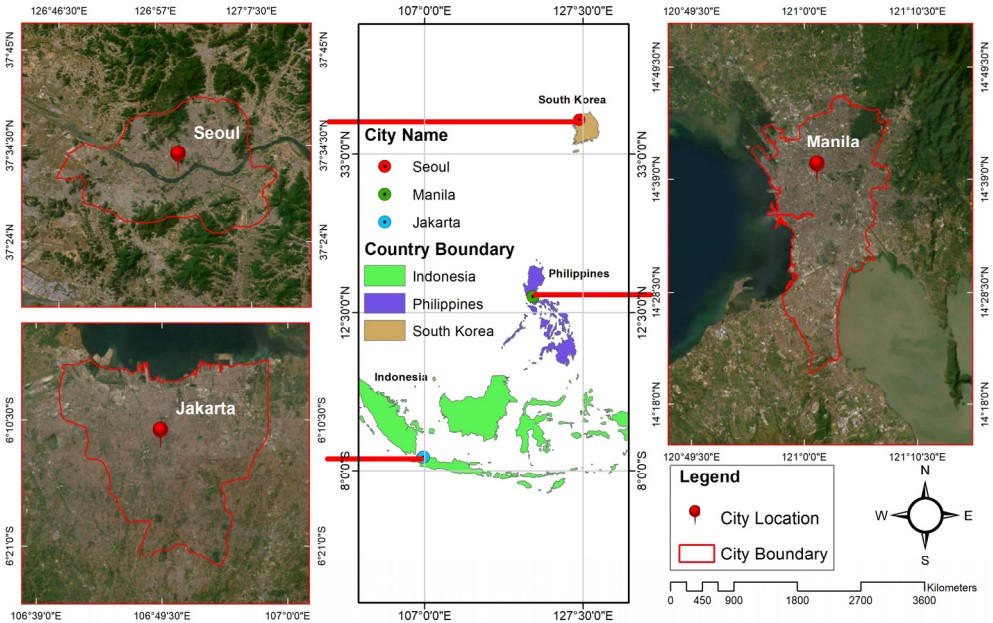

**Figure 2.** Selected cities from three different geographical locations.

### 2.3. Datasets

This study integrated SAR and optical satellite datasets to delineate urban impervious surfaces, utilizing both VV and VH polarizations from Sentinel-1 and modified indices from Landsat 8 OLI data. This integration was pivotal in overcoming challenges associated with land use classes, building shadows, and cloud cover, ultimately leading to enhanced accuracy in mapping urban impervious surfaces. Table 2 presents detailed information about the datasets and their corresponding bands used in this research. Moreover, in the

year 2023, a total of 37 satellite images were taken for Jakarta, 5 images for Manila, and 20 images for Seoul. The availability of a significant number of satellite scenes for Jakarta and Seoul within 12 months suggests a higher focus on monitoring and studying the region, possibly due to their large population and urbanization rate.

**Table 2.** Detailed information on datasets, bands, and polarization used.

| No. | Datasets | Bands/Polarization |
|-----|----------|-------------------|
| 1 | Sentinel-1 (SAR) | VV (vertical vertical) and VH (vertical horizontal) |
| 2 | Landsat 8 OLI (optical) | Band 2 (blue), Band 3 (green), Band 4 (red), Band 5 (NIR), Band 6 (SWIR 1), Band 7 (SWIR 2) |

*2.4. Processing*

This research used Landsat 8 multispectral imagery and Sentinel-1 SAR data. Landsat 8 provided six primary bands for generating normalized indices and calculating annual statistical compositions. Sentinel-1 SAR data contributed VV and VH polarizations, with textures generated using the grey level co-occurrence matrix (GLCM) technique in Google Earth Engine (GEE). Sampling datasets were generated using European Space Agency (ESA) and ESRI datasets, with points selected based on consensus regarding LULC classes [30]. These points were combined into a unified classification dataset. The dataset is then split into training and testing sets using an 80–20 percentile split. Before training the model, the forward stepwise selection (FSS) method used the 5-fold k validation method to define which feature and feature combination provides better accuracy. The training set of 80% data was used to train the model. The testing set of 20% data was used to calculate the model accuracy. The accuracy presented in the paper comes from the ground truth (GT) values. Concerning post-classification, an accuracy assessment was conducted by comparing model-generated classifications with reference datasets, providing insights to identify different land use and land cover types.

2.4.1. Preprocessing, Sampling, and Ground Truth Labels

Prior to the implementation of the machine learning classifier in this research study, a comprehensive preprocessing and dataset sampling procedure was conducted. This crucial phase involved the systematic preparation and transformation of the dataset to ensure optimal input for the subsequent machine learning model.

Three datasets, namely Dynamic World (DW) [25], European Space Agency (ESA) World Cover [31], and Environmental Systems Research Institute (ESRI) Land Cover [32], for the year 2021 were used. These three products have been carefully selected based on their significance and suitability for the present research. Details of all three selected products regarding dataset, model, and other characteristics are provided in Table 3.

These datasets were divided into four distinct classes: water, vegetation, bare soil, and urban impervious surfaces (UIS). Pixels where all three datasets agreed on the class were selected, and their corresponding classes were assigned. To generate a training dataset, random sample points were generated within the areas of 55 selected cities. For each generated sample point, the feature values and the agreed class from the three datasets were exported. This training dataset serves as the basis for further analysis and model development in the research paper. Furthermore, Table 4 provides a comprehensive overview of the three datasets, presenting detailed information regarding their respective classes and associated values. Subsequently, a systematic process was implemented to transform the initial class values into a new set of values. This process contributes to the clarity and consistency of the dataset, laying a foundation for accurate and meaningful interpretations in the context of the study. Because all three global products have different values for each class, the first step was taken to standardize their class code to a new code scheme from 0 to 6 (Table 4). Next, similar land cover classes were merged and assigned new codes from 0 to 4 based on a four-class scheme used for the present research.

**Table 3.** Detailed information on DW, ESA, and ESRI datasets.

| Dataset | Model | Methods | Features Name | Training Data | Input Label | Training Dataset Used for Training | Platform Used for Training |
|---|---|---|---|---|---|---|---|
| Dynamic World | DL | Fully Convolutional Neural Network (FCNN) | Blue, green, red, redEdge1, redEdge2, redEdge3, NIR, swir1, swir2 | 24,000 image (510 × 510 pixels) tiles were labeled by expert and non-expert group | 9 classes (water, trees, grass, flooded vegetation, crops, scrub/shrub, built area, bare ground, snow/ice) | Sentinel-2 L2A surface reflectance, NASA MCD12Q1; | GEE, Cloud AI |
| ESRI Land Cover | DL | Large UNET | 6 primary bands (red, green, blue, nir, swir1, swir2) | 24 k 5 km × 5 km image chips, all hand-labeled | 9 classes (water, trees, grass, flooded vegetation, crops, scrub/shrub, built area, bare ground, snow/ice) | Sentinel-2 L2A surface reflectance G6M5H6:J6 | Data access from Microsoft Planetary Computer and trained in Microsoft Azure Batch, AI-based |
| ESA World Cover | ML | CATBoost | 64 features are extracted from Sentinel-2, 12 features from Sentinel-1, 2 features from the DEMs, 23 positional features, and 14 meteorological features, for a total of 115 features | 10% of the points are sampled, with a maximum of 30 points per class per location | 11 classes (open water, trees, grassland, herbaceous wetland, cropland, shrubland, built-up, bare ground/sparse vegetation, snow/ice, mangroves, moss and lichen) | Sentinel-1, Sentinel-2, TerraClimate | GEE Terrascope with Python |

**Table 4.** Datasets used for sample generation, existing values, and recoding.

| DW | DW Value | ESA | ESA Value | ESRI | ESRI Value | Recode Value |
|---|---|---|---|---|---|---|
| Water | 0 | Permanent water bodies | 80 | Water | 1 | 0 |
| Trees | 1 | Tree cover | 10 | Trees | 2 | 1 |
| | | Mangroves | 95 | | | 1 |
| Bare | 7 | Bare/sparse vegetation | 60 | Bare ground | 8 | 2 |
| Grass | 2 | Shrubland | 20 | Rangeland | 11 | 3 |
| Shrub_and_Scrub | 5 | Grassland | 30 | | | 3 |
| | | Moss and lichen | 100 | | | 3 |
| Crops | 4 | Cropland | 40 | Crops | 5 | 4 |
| Flooded_Vegetation | 3 | Herbaceous wetland | 90 | Flooded vegetation | 4 | 1 |
| Built | 6 | Built-up | 50 | Built area | 7 | 5 |
| Snow_and_Ice | 8 | 7 | Snow and ice | 70 | 7 | 6 |

Where 0 = water, 1 = vegetation and flooded vegetation, 2 = bare soil, 3 = shrubland, grassland, and moss and lichen, 4 = cropland, 5 = built-up area, and 6 = snow/ice.

In a subsequent step, the snow/ice class was excluded from the analysis, and a revised classification scheme was implemented only for class for all three data products and for research results. Specifically, all types of vegetation, including vegetation, shrubland, grassland, moss and lichen, and cropland, were merged into a single category. The new classification system was simplified into four distinct categories: 0 for water, 1 for vegetation (encompassing shrubland, grassland, moss and lichen, and cropland), 2 for bare soil, and 3 for built-up areas.

During the feature selection process, 33,000 resample points (Appendix B) were extracted per class from the dataset, while allowing replacement. Therefore, the total dataset size became 132,000 points for four classes. The training dataset was split into 80/20 percentile as training and testing set. The training dataset was constructed using these resampled points. For the feature selection step using FSS, the training dataset was used, and a 5-fold cross-validation approach was implemented. This 5-fold cross-validation was performed inside the FSS method for better feature combination selection. This procedure enabled the selection of optimal features for further analysis without bias.

### 2.4.2. Random Forest Classifier

RF is a highly effective ensemble machine learning algorithm known for its ability to perform classification tasks [33]. It achieves this by constructing an ensemble of decision trees that are randomly selected from the training samples [34]. Random forest utilizes parallel processing and the bagging technique. Random forest, as a parallel classifier, constructs an ensemble of decision trees by independently training each tree on a different subset of the training data.

### 2.4.3. XGBoost Classifier

According to [35], XGBoost outperformed other classifiers in landslide susceptibility mapping. Ref. [36] also found XGBoost to be the most accurate predictor in their review of ensemble learning algorithms. XGBoost is a highly efficient machine learning algorithm for complex classification tasks and high-dimensional urban mapping.

One strength of XGBoost is its ability to handle noise and outliers, ensuring reliable performance. It also helps identify significant variables and provides insights into the underlying processes driving urban patterns through feature importance ranking. XG-

Boost effectively addresses imbalanced data issues using weighted loss functions and subsampling techniques. The scalability and speed of XGBoost make it well-suited for urban applications with large datasets and real-time processing requirements. In ensemble learning, XGBoost enhances prediction accuracy and mitigates overfitting, resulting in more robust and reliable results. In this study, the XGBoost classifier was implemented using equations outlined by [37], showing the working principle of the XGBoost. We have omitted the specific details of these equations, and the reference has been provided above.

### 2.4.4. Optical Temporal Indices

This study used a wide range of optical temporal indices derived from two band combinations of the six primary bands of Landsat (Table 2) and their four annual statistical composites (minimum, median, maximum, and standard deviation). From a pool of 60 spectral and temporal indices, VV and VH polarization, 122 Sentinel-1 GLCM textures, and bare soil-related indices are given in Appendix A. However, the study selected an optimized features list for random forest (RF) and XGBoost classifiers separately. For the selection process, the study uses the forward stepwise selection (FSS) method [38]. Therefore, based on FSS [39], eight features were selected for RF, and the first eleven features were selected for XGBoost.

The innovation of these modified indices is their utilization of annual statistical composites of each optical dual-band normalized indices. These modified indices incorporate both spectral and temporal information. By considering the temporal dimension of the data, these indices capture changes and patterns using multiple images for a single year, enabling a more comprehensive analysis of land use and land cover dynamics. The integration of spectral and temporal composite imagery enhances the accuracy and depth of the research analysis.

The nomenclature of optical features conveys three key pieces of information for each feature. Firstly, the abbreviation "ND" stands for normalized difference. Secondly, the combination of an uppercase and lowercase letter represents each corresponding band's first and last letter, with two bands denoted in this manner. Lastly, the final part of the feature name indicates the type of temporal composition undertaken. For instance, a feature like "ND_GnNr_Min" signifies that it involves normalized difference ("ND"), utilizing the green band ("Gn") and the near-infrared band ("Nr"), and employs a minimum operation for temporal composition. Finally, Min is short for minimum composition. All of these temporal compositions involve compositing all available images from the year 2023. The full list of optical features is given in Appendix A. Furthermore, details of all the optical features selected with some existing similar indices are illustrated in Table 5. Also, an additional NBWI novel index was implemented for the feature extraction process in this research.

**Table 5.** Description of optical temporal features and references of optical indices.

| Selected Optical Features | Similar Index | Reference |
| --- | --- | --- |
| ND_BeRd_SD | Normalized Pigment Chlorophyll Ratio Index (NPCRI) | [40] |
| ND_GnNr_Min, ND_GnNr_SD, ND_GnNr_Max | Green–Red Vegetation Index (GRVI) | [41] |
| ND_GnRd_Median | Visible Atmospheric Resistant Index (VARI) | [42] |
| ND_RdNr_Median | Normalized Difference Vegetation Index (NDVI) | [43] |
| ND_S1S2_Max, ND_S1S2_Median | Normalized Difference Tillage Index (NDTI) | [44] |
| NSAI1_median | Normalized Soil Area Index 1 (NSAI1) | [45] |
| swirSoil_median | swirSoil | [46] |
| ND_BeS2_Median | NBWI | A novel index was developed in this research |

Among all the optical temporal indices used in this study, a newly developed index, NBWI (Equation (1)), exhibits high reflectance in water areas to separate water from other land use classes.

$$\text{NBWI} = [(\text{Blue} - \text{SWIR2})/(\text{Blue} + \text{SWIR2})]_{\text{Median}} \tag{1}$$

The NBWI index is calculated by taking the difference between the reflectance values in the blue and shortwave infrared 2 (SWIR2) bands and dividing it by their sum. The index is typically applied to a median composite, which is created by taking the median value of multiple satellite images from the year 2023.

Although the NBWI is primarily a water-based index, it can still provide valuable insights in urban area studies. The NBWI helps in identifying and delineating water features within urban landscapes, assisting in mapping and understanding the distribution and extent of water bodies. This information is essential for urban planning, as it permits effective stormwater runoff assessment and mitigation strategies. By classifying water bodies, the NBWI also aids in assessing and managing flood-prone areas, optimizing drainage systems, and mitigating the effects of heavy rainfall events in urban areas.

Moreover, the NBWI contributes to studying the urban heat island effect. Water bodies have a cooling effect on their surroundings, and by accurately identifying and characterizing these features, researchers can analyze their influence on urban microclimates. This information can support strategies for urban heat island effect, urban green infrastructure planning, and enhancing the overall livability and sustainability of cities.

### 2.4.5. Textural Features

The authors utilized Sentinel-1 SAR data and the GLCM texturing function available in Google Earth Engine mentioned by [47]. These measurements provided insights into the randomness of grey-level distribution and the texture patterns present in the SAR data and contributed to the characterization and quantification of spatial structure and variability in the studied cities. The study designed each SAR polarization to be derived from a 10 m square ground spatial area. Each SAR texture feature name also shows three pieces of information. The naming started with polarization, the texture name, and the neighborhood length. For example, one such texture feature is "VV_dent_90m", based on a VV polarization-based texture of difference entropy with four neighborhoods. Similarly, all the SAR textures were generated, and a list is given in Appendix A.

The research methodology performance was evaluated using metrics such as overall accuracy, F1 score, precision, and recall.

### 2.5. Accuracy Assessment Method

The research proceeded to evaluate the accuracy of the developed framework by conducting an accuracy assessment task. A ground truth task was initiated, entailing the generation of 525 reference points for each city using the "create random point" tool in ArcMap. This process was conducted systematically, resulting in a cumulative total of 1600 reference points across all three cities. However, for each class, including water, vegetation, bare soil, and UIS, around 125 ground truth points were validated. Additionally, 25 points for each city were allocated for potential adjustments in the event of any identified errors.

The "Extract Values to Points" tool in ArcGIS Spatial Analyst (ArcGIS 10.8) obtained classified raster values at each reference point, which were exported to a CSV file. The accuracy assessment task was performed using Python code embedded in a Jupyter Notebook. This allowed for the execution of the necessary computations and facilitated the evaluation of model performance.

A random point sample was compared with ground truth points to ensure method performance, allowing a comprehensive evaluation of accuracy and reliability. The evaluation of classification model performance and key metrics was conducted based on the formulas outlined by [48]. Equations (2)–(5) were utilized to calculate overall accuracy, precision,

F1 score, and recall. These mathematical formulations provide a robust framework for assessing the performance of the classification model and determining important metrics. where

$$\text{Overall Accuracy (OA)} = (\text{TP} + \text{TN})/(\text{TN} + \text{FP} + \text{FN} + \text{TP}) \tag{2}$$

$$\text{Precision} = \frac{\text{TP}}{\text{TP} + \text{FP}} \tag{3}$$

$$\text{Recall} = \frac{\text{TP}}{\text{TP} + \text{FN}} \tag{4}$$

$$\text{F1 score} = \frac{2 * \text{TP}}{2 * \text{TP} + \text{FP} + \text{FN}} \tag{5}$$

i.   True positives (TP): number of samples correctly predicted as "positive".
ii.  False positives (FP): number of samples wrongly predicted as "positive".
iii. True negatives (TN): number of samples correctly predicted as "negative".
iv.  False negatives (FN): number of samples wrongly predicted as "negative".

## 3. Results

This section presents the research findings on extracting urban impervious surfaces with improved accuracy using the proposed fusion approach described in the methodology section. The results are further explained below, providing separate explanations for the UIS extraction process and other relevant parameters.

Research outcomes present the accuracy assessment of the results and compare them with the DW Google data product. The results indicate that the proposed framework outperformed DW regarding classification performance. The proposed approach achieved the highest overall accuracy of 81%, along with an F1 score of 78%, precision of 83%, and recall of 81%, indicating improved performance.

Table 6 provides a comparative accuracy assessment of the proposed framework with DW 2023. The evaluation used identical validation sample points for both datasets, ensuring a fair and consistent comparison. Table 7 presents the accuracy metrics, allowing for a comprehensive analysis of the performance of the proposed framework concerning DW.

**Table 6.** Selected features through forward stepwise selection (FSS) for different classifiers from Appendix A.

| Classifiers | Selected Features |
|---|---|
| Random Forest | "ND_GnNr_Min", "ND_GnRd_Median", "VV_dent_90m", "VV", "ND_RdNr_Median", "VH", "ND_GnNr_SD", and "ND_BeS2_Median" |
| XGBoost | "ND_GnNr_Min", "ND_GnRd_Median", "VH", "VV_dent_90m", "ND_S1S2_Max", "NSAI1_median", "ND_GnNr_Max", "swirSoil_median", "ND_S1S2_Median", "ND_BeRd_SD", and "VV_shade_90m" |

**Table 7.** Accuracy assessment values of DW dataset, RF, and XGBoost.

| Row Labels | Accuracy | F1 Score | Precision | Recall |
|---|---|---|---|---|
| DW | 0.763433333 | 0.733433 | 0.763167 | 0.763 |
| RF | 0.785633333 | 0.752867 | 0.784567 | 0.786 |
| XGB | 0.8109 | 0.7769 | 0.831033 | 0.811 |

Note: Colors shows the highest values of accuracy assessment metrics from red to green colors.

### 3.1. Confusion Matrix

The current framework's confusion matrix presents the predicted and true values for all three cities with two algorithms and one global product, as presented in Figure 3.

It provides a comprehensive overview of each class's accurate and inaccurate predictions. The confusion matrix is particularly useful in identifying classes that are being confused and misclassified by the model, thus providing insights into potential errors and misclassifications [49]. Figure 3 provides an evaluation of the classification accuracy across four different classes to showcase the ability of the proposed framework to classify samples correctly. It shows the collective performance for all three cities with four classes. Concerning all four classes and cities, the XGBoost classifier performed better compared to random forest.

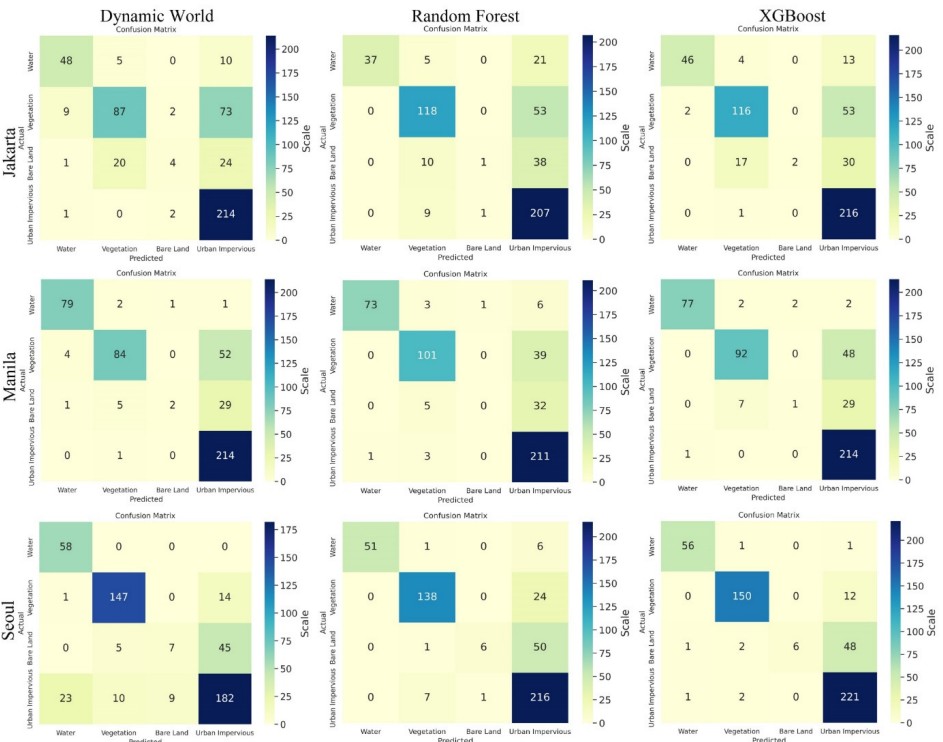

**Figure 3.** Confusion matrix: DW, RF, and XGBoost.

In Seoul, the DW dataset struggles to accurately identify positive samples, leading to lower true positives (TP). RF and XGB improve upon this, achieving higher TP values and overall better accuracy. XGB particularly excels in minimizing false positives (FP) and false negatives (FN), resulting in higher precision and recall. In Manila, the DW dataset faces difficulties in classifying samples as negative, resulting in lower true negatives (TN). RF and XGB again outperform DW, showcasing better precision and recall. XGB stands out with a balanced performance in identifying both positive and negative samples. For Jakarta city, the DW dataset struggles with both TP and TN counts, indicating challenges in identifying both positive and negative instances. RF and XGB consistently outperform DW, with XGB achieving the highest TP and TN counts, resulting in superior precision, recall, and overall accuracy.

Moreover, the analysis of confusion in Seoul showed both RF and XGB classifiers revealed higher accuracy, precision, recall, and F1 scores compared to the DW dataset. Particularly, XGBoost stands out with the highest values across all metrics, demonstrating its effectiveness in accurately classifying data points in the Seoul dataset. Similarly, in Manila, RF and XGB consistently outperform the DW dataset, with RF displaying a slightly higher accuracy, while XGB maintains competitive precision, recall, and F1 score values. The trend continues in Jakarta, where both RF and XGB classifiers demonstrate better performance than the DW dataset, with XGB achieving the highest accuracy, precision, recall, and F1 score.

The comparative analysis of confusion matrices underscores the enhanced classification capabilities of machine learning classifiers, specifically random forest and XGBoost, over a baseline dataset like DW. These results emphasize the importance of employing sophisticated algorithms for urban mapping tasks, with XGBoost emerging as the preferred choice due to consistently higher performance across multiple evaluation metrics in the given datasets from Seoul, Manila, and Jakarta.

### 3.2. Extraction of Urban Impervious Surface (UIS)

The methodology was implemented in three cities: Jakarta, Manila, and Seoul. The classified results of all three cities, representing four classes (urban impervious surface, water bodies, vegetation, and barren land), are presented in Figure 4. It is worth noting that although the method's primary objective was to extract urban impervious surface (UIS), the training process incorporated these main land use classes to ensure comprehensive training and classification accuracy.

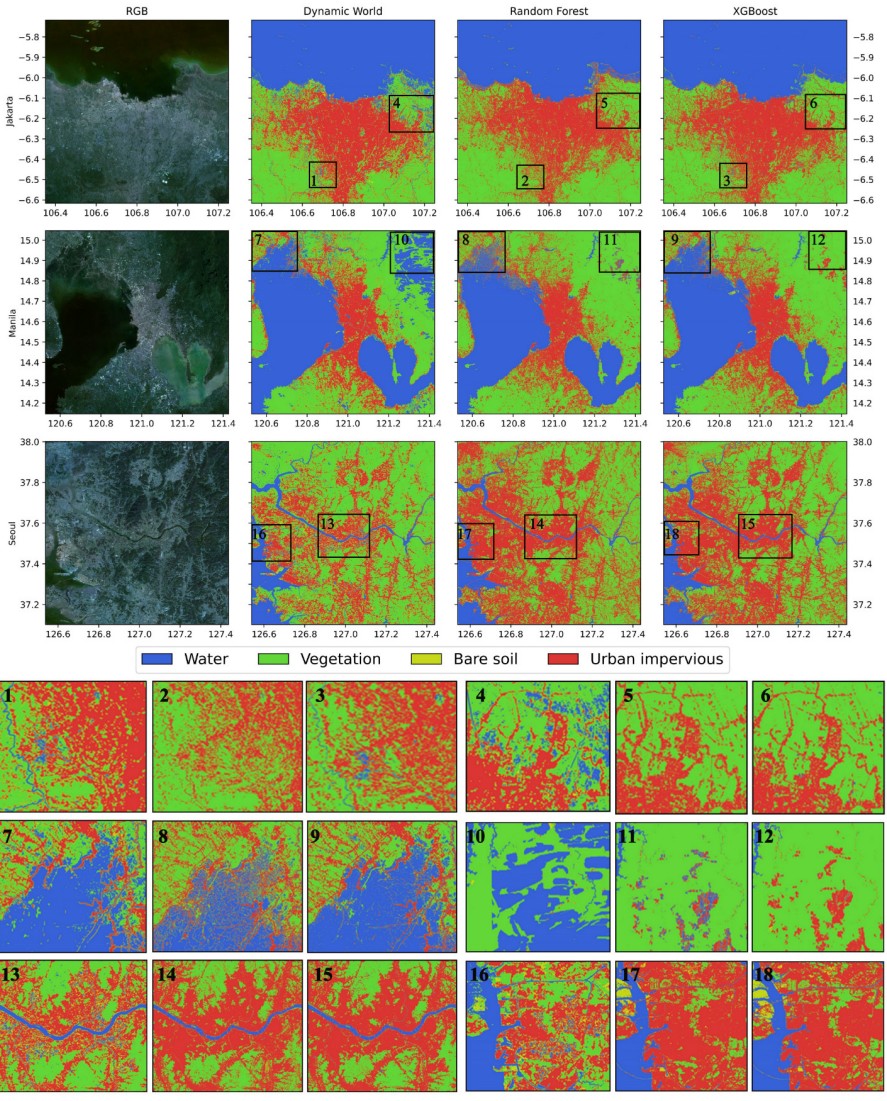

**Figure 4.** Classified map of selected cities using DW, RF, and XGBoost. Numbers show the same area for three different datasets (DW, ESRI, ESA).

The land use/land cover classification was conducted on selected cities using two different algorithms: random forest (RF) and extreme gradient boosting (XGBoost). The classification was based on the fusion of features extracted from Sentinel-1 (SAR) and

Landsat 8 (optical) datasets. The integration of modified indices and textures significantly improved the ability to identify urbanization patterns and map impervious surface areas.

The results showed that RF tended to underestimate the impervious surface (UIS) area across all cities, achieving an overall accuracy of 79%. On the other hand, the XGBoost classifier outperformed RF, achieving an overall accuracy of 81%. The research findings were not directly reached through the fusion of the two remote sensing datasets. Initially, the individual datasets were tested with both classifiers independently. The fusion of SAR and optical data through a stacked approach [50] discussed in Section 2, incorporating indices and textural features, improved classification results for all three cities.

This suggests that combining SAR and optical remote sensing data through a simple layer stacking fusion approach can yield more accurate land use/land cover mapping than relying solely on a single dataset. Fusing multiple data sources allows for synergistic utilization of their complementary information, enhancing classification performance. Therefore, the combined strengths of both Sentinel-1 SAR and Landsat 8 optical data, using the SLS fusion approach, provide a more comprehensive understanding of land cover characteristics, leading to improved accuracy in land use/land cover classification tasks.

*3.3. Comparison with Dynamic World Google Data Product*

The study outcomes were cross-validated using a reputable benchmark dataset, which provided an independent evaluation of the classification accuracy. Due to the unavailability of ESRI, ESA, and other well-known datasets with similar characteristics for 2023, the cross-validation was specifically conducted using the DW Google dataset alone. To ensure the reliability of the classification results, they were cross-checked and validated against the DW Google dataset. Dynamic World is a real-time dataset developed collaboratively by Google and the World Resources Institute, as described by [25]. This reference source is widely regarded as authoritative due to its rigorous production methods and stringent quality control measures.

A comparison of DW with the current research results is presented in (Figure 4) and corresponding accuracy metrics are shown in Table 7. The map shows three cities situated in diverse geographical settings, emphasizing the significant difference in UIS extraction. To emphasize the difference between the results obtained from DW and the current research findings, specific regions within all cities were magnified for specific regions. These highlighted areas from 1 to 18 (Figure 4) draw attention to the variations and discrepancies observed.

However, our approach demonstrated improved results in accurately delineating urban impervious surfaces at the city scale. Overall, research findings exhibit improved results compared to the other datasets, particularly in terms of distinguishing between classes and achieving higher accuracy in diverse cities.

**4. Discussion**

This research aims to improve the precision of urban impervious surface (UIS) extraction by combining optical and synthetic aperture radar (SAR) features. The study found that integrating these data types is more effective than using each dataset separately. Features such as texture and index, derived from the Landsat 8 surface reflectance dataset and Sentinel-1, were used. These features helped capture a mix of spectral, spatial, temporal, and scattering characteristics, enhancing the classification performance. A unique aspect of optical spectral indices is their annual composition, which gives them both spectral and temporal characteristics.

On the other hand, the textures of SAR polarization, which use varying neighborhood sizes, can capture spatial characteristics. Based on the fused features, the RF and XGBoost classifiers effectively modeled the complex relationships between land use classes. Accuracy assessment against ground truth and comparison with well-established global products validated the superiority of the proposed methods.

Many researchers fused two or more datasets, especially optical and SAR. For example, Ref. [22] achieved 75% accuracy using Sentinel-1 and Sentinel-2 fusion for land cover mapping in Colombia, but the authors implemented basic data from both sensors.

Another study conducted by [23] reported 76% accuracy for urban ecosystem mapping using Sentinel-1 and Sentinel-2 fusion. Similarly, research in the Eastern Brazilian Amazon [21] fused Sentinel-1 and Sentinel-2 and obtained 79% accuracy for land cover classification. As with the above studies, only the original band datasets were fused without additional derived inputs. However, the proposed approach of combining multiple feature types (indices, textures, and annual composites) extracted from Sentinel-1 and Landsat 8 provided improved classification performance over typical single or dual dataset fusion.

The proposed approach, which integrates multi-sensor open-source datasets, outperforms existing global dataset DW. The current method's claims of better results are supported by the higher accuracy of classification results compared to DW. This improvement can be attributed to the integration of the Sentinel-1 dataset with Landsat 8. Whereas DW only used the Sentinel-2 dataset, this inclusion of SAR data provided an advantage over DW. Furthermore, the temporally composited spectral index features were used. These composited spectral indices were derived from annually collected image collections; thus, capturing both spectral and temporal information enhanced the accuracy. However, unlike the deep learning (DL) model used by DW, simpler ML classifiers were selected but with a larger number of data points, which contributes to achieving higher accuracy.

However, it faces challenges in time series analysis due to the scarcity of historical images. It is often impracticable to acquire all datasets from the same historical year. While optical datasets are available through Landsat missions, the SAR Sentinel-1 dataset is not available before October 2014. Despite these limitations, the approach could enhance classification accuracy when applied to high-resolution datasets. Many data providers now offer preprocessed imagery, which is beneficial for analysis. However, despite the use of cloud removal algorithms in preprocessing, any remaining clouds can impact classification accuracy. Machine learning algorithms might even misclassify clouds or treat them as a separate class. Moreover, NBWI, a newly developed optical index, demonstrates high reflectance in water regions. This novel index holds promise for making substantial contributions to remote sensing applications, particularly in the identification and analysis of water bodies. One limitation identified in this study was the poor separability of the bare soil class compared to the ground truth data. This indicates that further improvements are needed to accurately classify bare soil in urban areas. Further research is still needed to improve distinguishing spectrally similar classes like bare soil and impervious surfaces for urban applications. Future research could explore alternative feature extraction methods or incorporate additional datasets to address this limitation and further improve the classification accuracy.

## 5. Conclusions

This research presents an optical–SAR data fusion approach for improved urban impervious surface extraction. Integrating texture, indices, and temporal features from optical and SAR datasets enables comprehensive land use classification. A unique aspect of this research is generating features through annual temporal composite features, rarely seen in other studies except for MaxNDVI. It can be concluded that using the XGBoost classifier outperforms RF and the existing Google data product (DW) for urban land cover classification. XGBoost achieves an overall accuracy of 81% and a precision score of 83%, demonstrating its superiority over RF and the Dynamic World global product. The utilization of multi-sensor data fusion in detailed urban mapping demonstrates effectiveness, offering improved accuracy in mapping impervious surfaces across diverse cities. Despite these advantages, the current approach encounters limitations, particularly in accurately classifying the bare soil category. It can be concluded that further enhancements are possible in the future to improve the separation of bare soil using the existing approach. This research contributes to the field of urban remote sensing by demonstrating the effectiveness

of ensemble machine learning classifiers and data fusion techniques for mapping urban impervious surfaces. The integration of optical and SAR features through the SLS technique, along with the use of the NBWI index, showed promising results in enhancing classification accuracy. However, the methodology can still contribute to sustainable urban planning and environmental management strategies by accounting for impervious surface dynamics under changing climatic conditions.

**Author Contributions:** Conceptualization, M.N.A.; methodology, A.J. and M.N.A.; validation, A.J. and M.N.A.; formal analysis, M.N.A.; data curation, A.J. and M.N.A.; writing—original draft preparation, M.N.A.; writing—review and editing, Z.S.; visualization, M.N.A.; supervision, Z.S.; funding acquisition, Z.S. All authors have read and agreed to the published version of the manuscript.

**Funding:** This research is supported by the National Key Research and Development Program of China with grant number SQ2023YFE0100956, in part by the Guangxi Science and Technology Program Guangxi Key R&D plan, GuiKe (2021AB30019), Sichuan Science and Technology Program (2022YFN0031, 2023YFN0022, and 2023YFS0381), and Hubei key R&D plan (2022BAA048).

**Data Availability Statement:** Data will be available on request.

**Conflicts of Interest:** The authors have no potential conflicts of interest.

## Appendix A. Features Collection for Forward Stepwise Selection

| Category | Features Names |
|---|---|
| Landsat dual-band normalized and temporal indices from six primary bands | "ND_BeGn_Min", "ND_BeGn_Median", "ND_BeGn_Max", "ND_BeGn_SD", "ND_BeRd_Min", "ND_BeRd_Median", "ND_BeRd_Max", "ND_BeRd_SD", "ND_BeNr_Min", "ND_BeNr_Median", "ND_BeNr_Max", "ND_BeNr_SD", "ND_BeS1_Min", "ND_BeS1_Median", "ND_BeS1_Max", "ND_BeS1_SD", "ND_BeS2_Min", "ND_BeS2_Median", "ND_BeS2_Max", "ND_BeS2_SD", "ND_GnRd_Min", "ND_GnRd_Median", "ND_GnRd_Max", "ND_GnRd_SD", "ND_GnNr_Min", "ND_GnNr_Median", "ND_GnNr_Max", "ND_GnNr_SD", "ND_GnS1_Min", "ND_GnS1_Median", "ND_GnS1_Max", "ND_GnS1_SD", "ND_GnS2_Min", "ND_GnS2_Median", "ND_GnS2_Max", "ND_GnS2_SD", "ND_RdNr_Min", "ND_RdNr_Median", "ND_RdNr_Max", "ND_RdNr_SD", "ND_RdS1_Min", "ND_RdS1_Median", "ND_RdS1_Max", "ND_RdS1_SD", "ND_RdS2_Min", "ND_RdS2_Median", "ND_RdS2_Max", "ND_RdS2_SD", "ND_NrS1_Min", "ND_NrS1_Median", "ND_NrS1_Max", "ND_NrS1_SD", "ND_NrS2_Min", "ND_NrS2_Median", "ND_NrS2_Max", "ND_NrS2_SD", "ND_S1S2_Min", "ND_S1S2_Median", "ND_S1S2_Max", and "ND_S1S2_SD" |
| Sentinel-1 VV and VH polarization after GLCM textures with Neighborhood Square Size | "VV", "VH", "VV_asm_30m", "VV_contrast_30m", "VV_corr_30m", "VV_var_30m", "VV_idm_30m", "VV_svar_30m", "VV_sent_30m", "VV_ent_30m", "VV_diss_30m", "VV_dvar_30m", "VV_dent_30m", "VV_imcorr1_30m", "VV_imcorr2_30m", "VV_inertia_30m", "VV_shade_30m", "VH_asm_30m", "VH_contrast_30m", "VH_corr_30m", "VH_var_30m", "VH_idm_30m", "VH_svar_30m", "VH_sent_30m", "VH_ent_30m", "VH_diss_30m", "VH_dvar_30m", "VH_dent_30m", "VH_imcorr1_30m", "VH_imcorr2_30m", "VH_inertia_30m", "VH_shade_30m", "VV_asm_50m", "VV_contrast_50m", "VV_corr_50m", "VV_var_50m", "VV_idm_50m", "VV_svar_50m", "VV_sent_50m", "VV_ent_50m", "VV_diss_50m", "VV_dvar_50m", "VV_dent_50m", "VV_imcorr1_50m", "VV_imcorr2_50m", "VV_inertia_50m", "VV_shade_50m", "VH_asm_50m", "VH_contrast_50m", "VH_corr_50m", "VH_var_50m", "VH_idm_50m", "VH_svar_50m", "VH_sent_50m", "VH_ent_50m", "VH_diss_50m", "VH_dvar_50m", "VH_dent_50m", "VH_imcorr1_50m", "VH_imcorr2_50m", "VH_inertia_50m", "VH_shade_50m", "VV_asm_70m", "VV_contrast_70m", "VV_corr_70m", "VV_var_70m", "VV_idm_70m", "VV_svar_70m", "VV_sent_70m", "VV_ent_70m", "VV_diss_70m", "VV_dvar_70m", "VV_dent_70m", "VV_imcorr1_70m", "VV_imcorr2_70m", "VV_inertia_70m", "VV_shade_70m", "VH_asm_70m", "VH_contrast_70m", "VH_corr_70m", "VH_var_70m", "VH_idm_70m", "VH_svar_70m", "VH_sent_70m", "VH_ent_70m", "VH_diss_70m", "VH_dvar_70m", "VH_dent_70m", "VH_imcorr1_70m", "VH_imcorr2_70m", "VH_inertia_70m", "VH_shade_70m", "VV_asm_90m", "VV_contrast_90m", "VV_corr_90m", "VV_var_90m", "VV_idm_90m", "VV_svar_90m", "VV_sent_90m", "VV_ent_90m", "VV_diss_90m", "VV_dvar_90m", "VV_dent_90m", "VV_imcorr1_90m", "VV_imcorr2_90m", "VV_inertia_90m", "VV_shade_90m", "VH_asm_90m", "VH_contrast_90m", "VH_corr_90m", "VH_var_90m", "VH_idm_90m", "VH_svar_90m", "VH_sent_90m", "VH_ent_90m", "VH_diss_90m", "VH_dvar_90m", "VH_dent_90m", "VH_imcorr1_90m", "VH_imcorr2_90m", "VH_inertia_90m", and "VH_shade_90m" |
| Soil-related indices (10) | "NSAI1_min", "NSAI1_median", "NSAI1_max", "NSAI2_min", "NSAI2_median", "NSAI2_max", "swirSoil_min", "swirSoil_median", "swirSoil_max", and "SISAI" |

**Appendix B. Features Collection for Forward Stepwise Selection**

| Dataset LULC Class | Value | Samples | Recoded LULC Class | Value | Samples | Training Class | Value | Samples |
|---|---|---|---|---|---|---|---|---|
| Water | 0 | 88,834 | Water | 0 | 88,834 | Water | 0 | 33,000 |
| Trees | 1 | 81,179 | Vegetation | 1 | 162,795 | Vegetation | 1 | 33,000 |
| Bare | 2 | 19,445 | Bare | 2 | 19,445 | Bare | 2 | 33,000 |
| Grassland | 3 | 28,176 | Built-up | 3 | 66,930 | Built-up | 3 | 33,000 |
| Crops | 4 | 53,440 | Snow/ice | 4 | 4341 | | | |
| Built-up | 5 | 66,930 | | | | | | |
| Snow/ice | 6 | 4341 | | | | | | |

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
