# Peer review of "Comparison of Random Forest and XGBoost Classifiers Using Integrated Optical and SAR Features for Mapping Urban Impervious Surface"

_remotesensing, doi:10.3390/rs16040665_

Round 1
Reviewer 1 Report
Comments and Suggestions for Authors
The manuscript “Comparison of Random Forest and XGBoost Classifiers using integrated Optical and SAR Features for Mapping Urban Impervious Surface” presents a data fusion approach to classification of urban impervious areas using Landsat and SAR (Sentential-1) data. The approach generates a large number of features from the two data sets including normalized differences (optical) and textural data (SAR). They then use a forward stepwise approach to limit the features for the final models. They compare XGBoost and Random Forest classifiers against a standard DW dataset from Google. They claim to use a novel normalized difference product they call NBWI.
I think there is some good work here, but it is very poorly presented and the most important aspects are not described or shown. This needs to be done before publication.
General Comments
The research seems interesting, but it is difficult to review because most of the important details are not included in the paper. The authors claim to have better accuracy than the Google DW data set, but they never describe how they generated their labled dataset for model training and accuracy assessment. The only statements are on L256-257 where they state “A third-party team 255 then performed ground validation using Google Earth Pro.” The ground truth (GT) is very important, I would like to see a much better description, some examples, specific pixels where DW was different from the GT. How many labeled samples were in each category, etc.
Next the models are not well defined. Was there a separate model for each city, one general model for all three, etc. I am not surprised that they can beat DW for urban areas, and their model is very specific. But it is impossible to review their work without more details.
Appendix A was not part of the review package. I assume this was just a list of the features initially used.
The manuscript mostly just makes general statements, many without citations. There is little detail on the actual work. How the ground truth were selected, what they looked like (i.e., numbers, distributions, etc.), more detail on the confusion matrix, and a discussion of classifications of the same area by different algorithms (Figure 4). For example, why is area 10 so different from areas 11 and 12, what do the features look like in this region, etc. Based on the RGB image, it appears the that DW gets the water correct in this region, then the paper algorithms miss it. It would be nice to see detailed RGB to confirm.
The paper needs more analysis of the methods, where they work, where they fail, etc. For Figure 4, the RGB image needs a gamma correction to make it brighter. It would also be good to have RGB images of the detail areas.
For the labeled (e.g., ground truth) data it would be good to have a map of the pixels that were selected, how they were distributed among the cities and the types of ground cover.
Specific Comments (manuscript markup attached)
L15 What “novel indices”? Need to list (and in the paper define why they are novel.
L23 “Higher accuracy” than what? I think this is an sentence construction problem.
L33 Change “high” to “higher”
L40 need citation
L41 need citation
L45-47 need citations
L49 Change “O-S (Optical SAR)” to “Optical SAR (OS)”
L51-58 need citations
L59-63 need citations
L66 “overall accuracy” for what? Or compared to what?
L68 Same questions about SVM accuracy. What is being classified?
L78 SLS already defined – don’t need to redefine.
L88-94 need citations
L99 SLS already defined – don’t need to redefine.
L103 NBWI need equations and why it is novel. Normalized band indexes are very common. We have a standard function that generates normalized band indexes for all the Landsat band pairs. What is novel about NBWI?
L117 What is the advantage of your approach to just using DW? i.e., if SW already has this information why do we need a new algorithm (yours is more accurate – need to state the reasons you tried to develop a new one).
L129 Tabel 1 formatting
L135 Seoul is NOT Tropical, by the Koppen-Geiger classification it is “Dwa” which is “humid continental with dry winters”
L147 Figure 2 impossible to read text – figures too small
L151-154 The statement about how the algorithm overcomes various issues is a ‘result’ and needs to be later in the paper – probably the discussion section or maybe in the results section.
L158 Table 2 formatting 1st column. Also, don’t need GEE data links – just the name of the data set.
L167 Define GLCM on first use
L168-170 need citations
L168 What does “datasets were refined using…” mean? You need to provide detailed descriptions of how your ground truth data were generated, with examples, statistics, and methods. This is a major problem with the current manuscript.
L173-174 How were the training/ground truth data labled, provide detailed method, provide stats, provide examples. The entire paper is based on these ground truth data and they are not presented.
L198-199 You don’t need the equations for XGBoost. Just cite the relevant papers unless you modified the equations somehow.
L201 Reference not found.
L205 Appendix A not provided.
L226 Appendix A not provided
L227 Define NBWI use equations, show why important. Why is it better than other normalized band differences? Rarely is classifying water an issue in urban classification.
L231 Should define NBWI acroynim on first use, not here.
L231-232 Do you have data or a citation to show that NBWI is better on open ocean water – need either data or citation. Also, once again what relevance does this have for urban land use classification?
L249 Need to describe ground truth used for error metrics.
L255-256 How did the “third party team” provide ground truth? Who was the third party team? How was the third party team selected? Why was a third party team used? You need to clearly define the methods used to generate ground truth.
L260 Need to describe what “A random point sample was compared with ground truth points” means? Why wouldn’t you compare to all the ground truth points? How many random points were used? What land cover classes were included in the random points? How many points in each classification? Etc. The biggest question is why you didn’t use all the ground truth points?
L274-279 formatting
L307 Figure 3, need to discuss and explain, not just present the figure.
L307 Figure 3, impossible to read axis labels or numbers.
L315 Delete the phase “characterized by heterogeneous extents” or explain what this means – I have no idea and it isn’t needed.
L318 Define “UIS”
L326-329 Here you talk about models not using the fused data – this needs to be presented and discussed in the paper – show how fusion is better, etc. (or cite the research where this was done)
L342 Figure 4 I suggest just having the 6 areas numbered (with the same area – then have columns by model type. It took me several minutes to realize I was supposed to compare areas 10, 11, and 12. Need detailed plots at a level to see individual pixels. Need RGB detailed plots at the same scale.
L342 Figure 4 Need a discussion of this figure – this is the heart of the whole paper and it is not discussed. For example, why does DW have so much more water in area 10 that your models? Which is correct? Show a detailed RGB so we can tell
L342 Figure 4 Need to work on the RGB image be visual analysis- brighten using gamma correction, maybe saturate the colors for the scale of the image, etc.
L359-360 Need to actually discussion Figure 4, where are there differences, where is there agreement, what causes problems for the different models. Figure 4 is the heart of this paper and is not discussed or described at all. Most of the text is just general hand-waving that could have been written before you had results.
L390-391 There were no data about the NBWI, why it is novel, how it is new, how it is better. You either need to provide data to back up these claims or a citation. Also, NBWI is mentioned a number of times but you don’t show how it helps your models. The only statement is that it is good in the open ocean which doesn’t matter for urban land classification.
L390-391 and L229 Table 2. NBWI is the last feature listed in Table 2. Does this mean it had the least importance in the stepwise regression? Also, stepwise regression depends on the order in which the features are presented to the model. How were they presented? If they were presented in a different order would you have had different features selected? What was the importance of each feature? There are only 11 features, you could provide the weights used in the model and the cut-off values for the different classes.

English is good. There are a few awkward sentences or phases.
Fine for publication as is.
Author Response
Thank you sincerely for your constructive suggestions, comments, and the valuable time you invested in reviewing our manuscript. And facilitate us in revising and improving our submission. We have carefully incorporated all the suggested changes into the revised manuscript file, and responses to each comment are provided in the attached file.
Please see the attachment where both the revised manuscript and response to your comments are combined into one file and attached below.

Reviewer 2 Report
Comments and Suggestions for Authors
1. Lines 102 to 105, "For Landsat 8 data, a novel index, the Normalized Blue Water Index (NBWI) and modified indices such as the Visible Atmospherically Resistant Index (VARI) and the Normalized Difference Built-up Index (NDBI) were implemented, utilizing all available images for the selected year (2023). " Please provide the basis and reasons for constructing this novel index and modified indices.
2. The Datasets section can include more detailed information, such as how many scenes were used or which months of imagery.
3. “Error! Reference source not found.” in lines 201 to 202. What does it mean?
4. "This study employs the integration of SAR and optical satellite data to define urban superior surfaces, utilizing both VV and VH polarization from Sentinel-1 and Landsat-8 OLI data." This will lead to misunderstandings, as VV and VH polarization also include optical images.
5. Shouldn't the results of Accuracy Assessment and Fusion Matrix be placed in the Results section? Why is it placed in the Materials and Methods section.
6. Both the abstract and conclusion sections of this article indicate that the newly constructed NBWI shows promising results in improving the accuracy of extracting urban impermeable surfaces. But in the results and discussion section of the article, I did not see the importance and degree of influence of NBWI in the classification process. I suggest adding more analysis of NBWI in the article to demonstrate the significance of constructing this novel index.
Comments on the Quality of English LanguageMinor editing of English language required
Reviewer 3 Report
Comments and Suggestions for Authors
Dear Authors,
I have read your text carefully and want to congratulate you on your interesting research experiment. I find the overall article structure to be appropriate. I assess your manuscript as informative and credible. However, I noticed some minor shortcomings needing improvements:
- some references are incorrectly cited (e.g., lines 49, 178, 207, etc.),
- line 126 - doubled space (please see the whole text because there are some more similar errors),
- table 1 - error in column 1,
- Figure 2 - the scale bar in kilometers is wrongly divided (only round values are allowed, not just random ones),
- Figure 2 - the middle chart is difficult to read; different colors of dots representing particular cities are not reflected in the neighboring pictures (only red dots are visible),
- Table 2 - errors in column 1,
- line 201 - an error has appeared,
- Line 229 - there is again Table 2 - wrong numbering;
- Table 5 - challenging to interpret according to colors - the highest values should be shown as red going down to the lowest ones, which should be marked green - not reversely.
Summarizing - please go through the text again, trying to improve it with careful attention to details.
Good luck!
Reviewer 4 Report
Comments and Suggestions for Authors
The research conducts a comparative analysis of two ensemble machine learning classifiers, namely Random Forest and Extreme Gradient Boost, leveraging the integration of optical and SAR features alongside simple layer stacking techniques for extracting impervious surfaces in urban areas. The study is centered on three East Asian cities: Jakarta, Manila, and Seoul, each characterized by unique urban dynamics. These cities exhibit distinct characteristics regarding climate, population densities, and urbanization rates.
A novel index, termed the Normalized Blue Water Index, is introduced in this study to effectively distinguish water from other features, serving as a valuable optical feature. Extreme Gradient Boost demonstrated superior classification accuracy compared to Random Forest and Dynamic World, underscoring its potential for urban remote sensing applications.
There is a file attached with corrections. It is necessary to discuss the results obtained in comparison with other authors

Round 2
Reviewer 1 Report
Comments and Suggestions for Authors
The manuscript “Comparison of Random Forest and XGBoost Classifiers using integrated Optical and SAR Features for Mapping Urban Impervious Surface” presents a data fusion approach to classification of urban impervious areas using Landsat and SAR (Sentential-1) data. The approach generates a large number of features from the two data sets including normalized differences (optical) and textural data (SAR). They then use a forward stepwise approach to limit the features for the final models. They compare XGBoost and Random Forest classifiers against a standard DW dataset from Google. They claim to use a novel normalized difference product they call NBWI.
I think there is some good work here, but it is very poorly presented and the most important aspects are still not described or shown. This needs to be done before publication.
General Comments
The research seems interesting, but it is difficult to review because most of the important details are not included in the paper. The authors claim to have better accuracy than the Google DW data set. But the DW dataset was used for both training and ground truth (as near as I can tell).
Based on my understanding of the current manuscript.
· They relabeled land classifications from DW ESA and ESRI to either 5, or 6, or 4 classes (all these statements are made).
· They used random points to generate 33,000 training/testing points. We don’t know how many points in each category or anything else,
· they trained models (I think one per city, but this isn’t stated).
o If it were one per city (the most likely) how many points were there per city, did each get 11,000?
· Then they generated 1,600 “reference” points. I think these were used for ground truth to determine accuracy, but it isn’t clear. It also isn’t clear from which data set these points came from, I assume from the fused landcover, relabeled data set. Again we don’t know how many points in each class, each city, etc.
This all needs to be clearly laid out. What data were used, what was done, how was it evaluated.
Since they use the DW data set as ground truth (it seems, though it isn’t clear – it seems that they combined several data sets, and only included points where all the datasets agreed – though that also isn’t clear), it isn’t clear how they can state they are more accurate than DW, and DW should agree with the groundtruth. As per their statement, only where classes agreed were they used.
L183-L184 This again seems to be a basis for confusion, how were things transformed, what was transformed, what was the “systematic process”, how many points was it applied to? How did this change the DW values – this might be where the difference in the models comes from.
This question from the original review was not answered in the manuscript.
“Next the models are not well defined. Was there a separate model for each city, one general model for all three, etc.”
The response to comments just says that it is more clearly presented. I still want to know if you trained a model for each city or one general model for all three. The response says “the same methodology was used” I think this means a model for each city, but this is never stated or explained in the manuscript (or the cover letter).
I’m sure there are other issues, but until I can understand what data were used to train the models, what models were created, and how were ground truth data generated, the rest of the paper doesn’t matter.
Last comment, they claim better accuracy than DW, but don’t explain how they quantified the DW accuracy. What was the ground truth, what was the confusion matrix, etc.
Some specific comments
L173-186, Need to spell out data set acronyms. Need to define what these data sets are? Did all three datasets have the same land cover classification? If not, how were they divided ton the 5 classes used in the manuscript? What date were the land classification data sets from, what dates were the images from? Were there changes between times, etc. Table 3 presents some of this information, but it needs to be described and discussed in the text.
L174 & L193 Did you use 5 classes or 4 classes, L174 says 5, L193 says 4. Table 3 has 8 classes.
L199 5-fold cross validation used for feature selection. This brief statement is the research, it is never described. Again – you need to describe exactly what you did, what the values were, etc.
L271-276 Why are you discussing Urban Heat Island effects? They aren’t a part of this study. You don’t demonstrate that NBWI is better for them than NDWI. If you don’t have data, don’t make these statements.
L196 – L296-L301 How do the 1600 reference points relate to the 33,000 sample points? Were any the same, were they close? Why a separate selection, rather than just a train/test split on the 33,000?
L307 What is the difference between the random point sample and the ground truth points? Is one a subset of the other? Reading this, I still don’t know what you did.
Comments on the Quality of English Language
English is generally good. There are some typos.
Author Response
We have extensively added the detail which were making confusion regarding, Sampling, how sample were collected, training/testing datasets, and ground truth. Moreover, all these points were incorporated in manuscript file. Please see attached file for point to point response

Reviewer 3 Report
Comments and Suggestions for Authors
Dear Authors,
Thank you very much for the improved version of your article. I have read your responses to my comments, and I appreciate them entirely. Hence, as I don't have any other objections, I recommend publishing your article.
I wish you good luck with your further research!
Author Response
Thank you very much for appreciating the author's response to your comments.
Best wishes to you as well.

Reviewer 4 Report
Comments and Suggestions for Authors
Thanks for all the corrections
Author Response

(The authors gave the same response as above.)
